# A Domain-Driven Framework to Analyze Learning Dynamics in MOOCs through Event Abstraction

Luciano Hidalgo [ID] and Jorge Munoz-Gama *[ID]

Department of Computer Science, Pontificia Universidad Católica de Chile, Santiago 8331150, Chile
* Correspondence: jmun@uc.cl

**Abstract:** Interest in studying Massive Online Open Courses (MOOC) learners' sessions has grown as a result of the retention and completion issues that these courses present. Applying process mining to study this phenomenon is difficult due to the freedom of navigation that these courses give their students. The goal of this research is to provide a domain-driven top-down method that enables educators who are unfamiliar with data and process analytics to search for a set of preset high-level concepts in their own MOOC data, hence simplifying the use of typical process mining techniques. This is accomplished by defining a three-stage process that generates a low-level event log from a minimum data model and then abstracts it to a high-level event log with seven possible learning dynamics that a student may perform in a session. By examining the actions of students who successfully completed a Coursera introductory programming course, the framework was tested. As a consequence, patterns in the repetition of content and assessments were described; it was discovered that students' willingness to evaluate themselves increases as they advance through the course; and four distinct session types were characterized via clustering. This study shows the potential of employing event abstraction strategies to gain relevant insights from educational data.

**Keywords:** event abstraction; MOOC; learning dynamics; process mining; in-session behavior

## 1. Introduction

Massive Open Online Courses (MOOCs) are now an option for students from all over the world who seek to learn at their own pace and with flexible deadlines from a wide range of courses and programs [1]. Since they can scale seamlessly to hundreds or thousands of students [2] and are commonly free for people who have no intention of getting a certificate, interest in these courses continues to grow [3]. Despite this, MOOCs face significant challenges, such as high attrition, low enrollment of users from developing countries, and poor completion rates [1]. Although these problems have afflicted MOOCs since their inception, interest in understanding student behavior through research on phenomena such as dropout, motivation, and self-regulated learning has increased in recent years [3]. This has led to the study of more specific behaviors, such as actions [4], strategies [5], tactics [6], sessions [2] and, temporal dynamics [7]. In particular, the study of learning dynamics within a session, that is, during a period of uninterrupted work, is receiving increasing attention [2,8].

The field of *Process Mining* [9–11] has been viewed as a promising tool for answering research questions in educational settings due to its ease of use for users who are not necessarily experts in data mining or process science [5,12]. Process mining algorithms are able to automatically discover (or mine) a model representing the dynamic behavior of end-to-end processes based directly on event data. In the literature, a common approach is to use columns directly from a database table as activities (i.e., steps of the process) for process mining algorithms, such as the accessed MOOC resource [12]. Due to the freedom students have to navigate through the resources, they often generate "spaghetti models", as shown in Figure 1. To overcome these scenarios, most authors end up tinkering with

process discovery tools, applying filters to routes and activities, or altering activity and case ID until they obtain a readable, albeit incomplete, model, as depicted in Figure 2. This narrows the scope of questions that can be answered using these techniques.

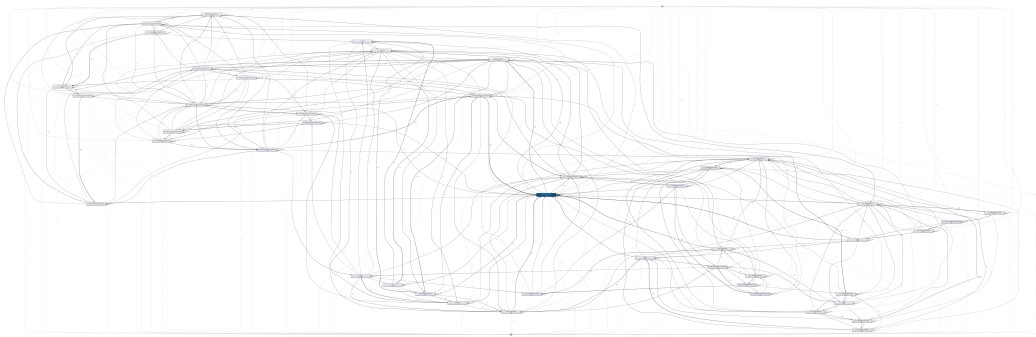

**Figure 1.** Conventional unfiltered spaghetti model obtained from raw data. The model is generated in Disco using data from an introductory Python programming course, with the resource name field serving as the activity and the student ID serving as the case ID. The source data is the same as that used in the case study section.

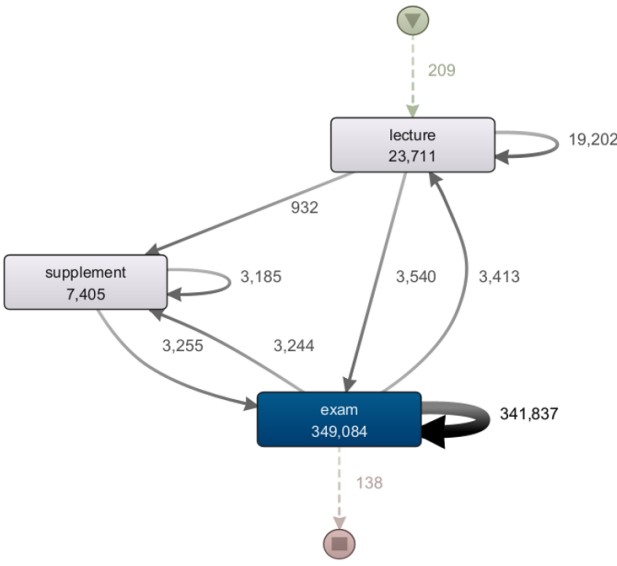

**Figure 2.** Oversimplification of the previous model, using the same data but this time with the resource type field as activity.

This raw, unfiltered data does not fit more complex questions, so we need techniques that allow us to group low-level events into high-level activities, i.e., event abstractions. There are numerous event abstraction methods described in process mining literature (for a review, see [13]). Most event abstraction approaches are data-driven (bottom-up), that is, they use domain-agnostic or unsupervised methods to detect frequent patterns in data. However, in some circumstances, high-level activities have already been specified in a domain-driven (top-down) manner, and the objective is to find those activities in the data. Such approaches are not appropriate in these cases. For example, the same behavior of accessing a MOOC resource may indicate if a learner is studying from it or just scanning it to understand what will come next in a lesson. Finally, constructing an event abstraction might be a difficult task for educational decision-makers who are not specialists in process mining. This is due to the fact that it requires a complete understanding of concepts such as case ID, activity ID, and event ID. As a result, easy-to-follow recipes must be defined in order to apply process mining in interdisciplinary contexts such as education.

We propose a domain-driven event abstraction framework to analyze learning dynamics in MOOC sessions, involving three steps: (1) a minimal data model that can be

adapted to most MOOC systems; (2) the definition of a low-level event log, including the definition of case ID and user sessions; and (3) the definition of seven high-level activities, our learning dynamics, and its corresponding high-level event log. The framework maps all low-level events to the seven learning dynamics discussed here, making it straightforward enough for educational managers to replicate. After which, this event log can be analyzed using traditional process mining methods or machine learning approaches to gain insights into the dynamics of students in MOOC sessions. Furthermore, the framework is adaptable enough to be used for courses that include other types of content, such as summaries or *cheat sheets*, self-reflection activities, formative assessments, projects, and forums. We validate and demonstrate the framework's use with a case study of sessions from students who successfully completed an introductory programming course on the Coursera platform (Figure 3). To that end, we propose three research questions on how students interact throughout Coursera course sessions:

- **RQ1**: *What are the characteristics of the sessions that involve learning dynamics in which a resource is revisited?*
- **RQ2**: *Are there differences in terms of learning dynamics between the first and final sessions carried out by students?*
- **RQ3**: *What types of MOOC sessions do successful students go through and how do they differ from each other?*

Although these are the questions we employed for the case study, our framework is flexible enough to handle different ones as long as they fit within the context of study sessions.

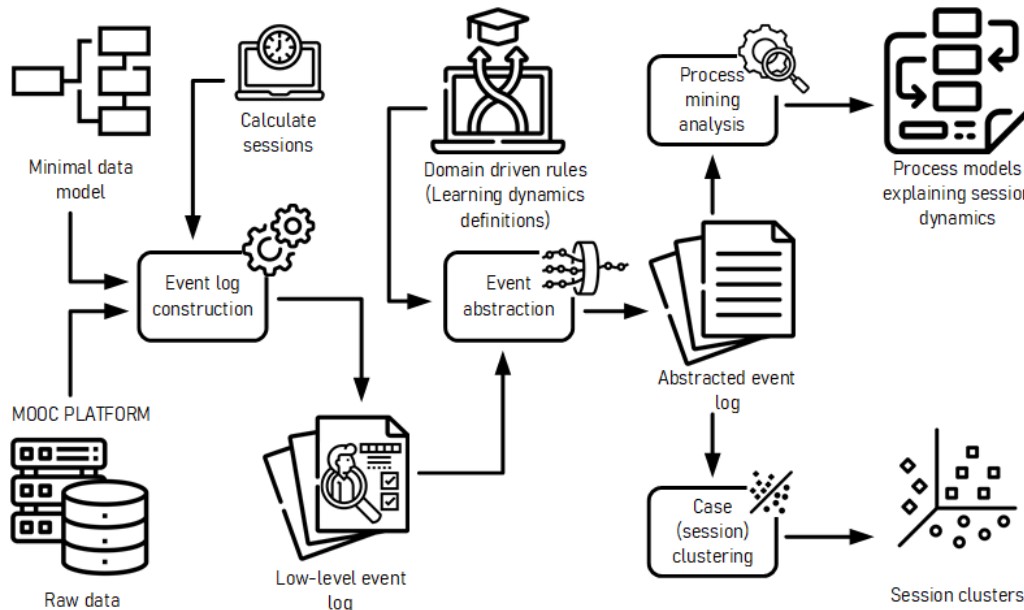

**Figure 3.** Application of the proposed framework to Coursera course data. First, a low-level event log is constructed using the minimal data model and raw data. Then, the learning dynamics are applied in order to abstract the activity sequences into a high-level event log. Finally, this event log is used to answer the research questions through process mining and clustering.

This research builds on a preliminary workshop paper published in [14]. In particular, among the additions, we highlight (1) a comparison of the proposed learning dynamics with what is known from research, (2) an expanded explanation of the framework, (3) an analysis of the abstracted study sessions applying clustering, and (4) an extended discussion based on the most recent works published on the subject.

The remainder of the paper is organized as follows: Section 2 discusses related works in process mining in education as well as event abstraction. Section 3 presents the framework's stages and the proposed learning dynamics for constructing an abstracted high-level event log. Section 4 outlines the clustering process that was used to analyze the sessions that

resulted from the case study application. Section 5 describes the case study that was used to demonstrate the framework's application and evaluate the results obtained from it. The dataset used and the process of applying the framework are detailed here, as well as an analysis of the research questions and their corresponding outcomes and a discussion of what was achieved. Finally, in Section 6, we summarize our results and limitations, as well as future research directions in the topic.

## 2. Related Work

This section discusses work related to both MOOCs and event abstractions in the context of process mining.

### 2.1. Process Mining and MOOCs

Although MOOC systems generate a large amount of data, research on process mining approaches is still in its early stages [12]. Several authors, however, have attempted to describe or investigate student processes using this data. One of the first works on the subject, [15], uses process discovery and conformance techniques to categorize behavioral differences among groups of students. More recent studies such as [7] assess the behavioral differences between students who both complete and do not complete a MOOC from the specific perspective of time commitment. In the same context, [16] investigates the differences in the process between three different sets of students depending on whether they have completed all, some, or none of the MOOC activities. [17] compares the use of process mining and sequence mining to predict student dropout from a MOOC, while process mining provides useful tools for analysis, sequence mining provides better predictive results. Dotted charts, fuzzy miner, and social miner are used in [18] to establish differences between groups of students with high and low performance.

By combining clustering approaches with process mining, [19] identifies four sets of students, ranging from those who drop out at the very beginning of the course to those who successfully complete it. Their research shows that students in the cluster of individuals who successfully completed the course tended to watch videos in successive batches. Furthermore, using clustering techniques and process mining, [20] proposes an approach to measuring the difficulty and importance of videos in a MOOC.

[5] examines the event logs of three MOOC Coursera courses and discovers six patterns of student interaction. According to the behavior described, these patterns were also classified into three clusters: sampling learners, comprehensive learners, and targeting learners. These findings are expanded and refined in [21]. Furthermore, the idea of "session" has been used as a unit of analysis in both studies. In the same regard, [2] investigates the behavior of students in work sessions in greater depth based on eight different possible interactions, segmenting them into those who complete and those who do not complete the course. Their research found that students who complete the course exhibit more dedicated behavior and attend a greater number of sessions during the course. [6], like our study, aims to uncover high-level patterns in low-level events. Their research looks at student behavior in MOOCs via the lens of nine previously identified learning actions [4], which are subsequently utilized to find higher-level patterns known as learning tactics.

### 2.2. Process Mining and Event Abstraction

Regardless of how useful process mining techniques are for understanding how organizations work, the systems that generate this data do not always do so at an adequate or consistent granularity level [22]. As a result, techniques that enable the abstraction of high-level activities from granular data are critical for the proper application of process mining [13]. There are now various strategies in place to solve this issue. Unsupervised machine learning is used in one approach family to group events based on several dimensions, such as the semantics of activity names [23], the physical proximity in which events occur [24], events that occur frequently together [25], and sub-sequences of activities that are repeated [26], among others. Other authors have proposed less automated strategies,

for example, [27], who groups elements according to the relationships between entities (ontologies) in order to abstract events using domain knowledge, which was successfully applied in the medical domain. Similarly, [28] provides a four-stage strategy based on prior identification of process activities, granular matching of activities and events based on their kind, and context-sensitive criteria. Indeed, this concept suggests categorizing various events as activities. Furthermore, other authors combine supervised and unsupervised techniques to present an event abstraction strategy in diffuse contexts [29]. As a result, the aforementioned approach is based on the division of events into sessions based on activity periods prior to the formation of clusters of events, which are then manually reviewed in a heat map and mapped to high-level activities.

### 3. Domain-Driven Event Abstraction Framework

This section introduces the domain-driven event abstraction framework, which aims to assist educational managers in creating a high-level event log to analyze students' learning dynamics throughout MOOC work sessions. The framework is composed of three stages: (1) a minimal data model capable of being mapped to any MOOC system; (2) the definition of a low-level log based on the minimal data model; and (3) the construction of a high-level log derived from the low-level log.

### 3.1. Stage 1: Minimal Data Model

The minimum data model is the first stage that specifies the framework. This is a data model that contains only the information required to construct the low-level event log, which acts as a common ground throughout various MOOC systems, including Coursera, FutureLearn, and edX, among others. Figure 4 depicts the minimal data model, which is populated with data each time a user interacts with a MOOC resource. The model requires identifying three major elements: the resource dealt with, the user who performs the interaction, and the moment at which the interaction is made. All MOOC systems use a unique identifier to differentiate resources and users. It is also necessary to identify the order in which the resources in the course are organized. This makes it possible to determine whether the user is engaging with the resources in a sequential or chaotic manner. The model also specifies the type of resource under consideration. This proposal defines two generic types: content resources (video lectures, presentations, etc.) and assessments (quizzes, exams, etc.). The framework is easily extensible to add other sorts of resources, such as projects or forums. Finally, in the event log, each interaction with a resource is assigned a state (start or complete). This allows us to determine if students' learning dynamics correlate with exploratory or in-depth work patterns. The majority of MOOC systems contain the information required to assess status. In certain circumstances, such as Coursera, the state is explicitly recorded in the Course Progress table as two distinct interactions (one marked "Started" and the other "Completed"). Other systems can identify status based on two timestamps ("Start" and "End") linked to the same resource.

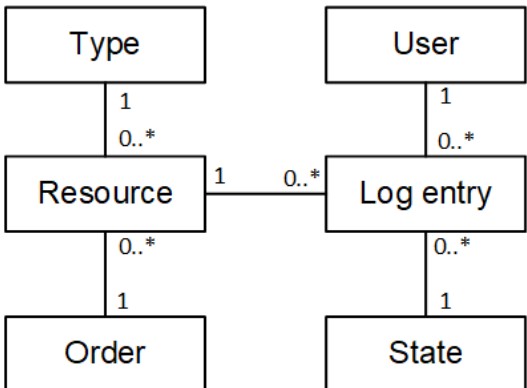

**Figure 4.** Minimal data model suggested.

### 3.2. Stage 2: Low-Level Log

The framework's second step shows how to construct the low-level log using the information given in the previous stage's minimum data model. Each interaction with a MOOC resource recorded in the basic data model represents an event in the low-level log, as shown in the example in Table 1. With the exception of two elements, session ID and case ID, the transition of the data in the minimum data model to the low-level log is straightforward.

**Table 1.** Example of an event log fragment.

| User | Resource | Timestamp | Order | Type | State | Session |
|------|----------|-----------|-------|------|-------|---------|
| Johnny | Guitar Basics: Guitar parts | 2020/05/28 13:05:10 | 1 | Content | Started | 1 |
| Johnny | Guitar Basics: Posture | 2020/05/28 13:05:18 | 2 | Content | Started | 1 |
| Johnny | Guitar Basics: Picking | 2020/05/28 13:06:02 | 3 | Content | Started | 1 |
| Johnny | Guitar basics: Identifying notes | 2020/05/28 13:06:30 | 4 | Content | Started | 1 |
| Johnny | Guitar Basics: Guitar parts | 2020/05/30 12:30:22 | 1 | Content | Started | 2 |
| Johnny | Guitar Basics: Guitar parts | 2020/05/30 12:33:01 | 1 | Content | Completed | 2 |
| Johnny | Guitar Basics: Posture | 2020/05/30 12:35:18 | 2 | Content | Started | 2 |
| Johnny | Guitar Basics: Posture | 2020/05/30 12:37:55 | 2 | Content | Completed | 2 |
| Johnny | Guitar Basics: Picking | 2020/05/30 12:38:25 | 3 | Content | Started | 2 |
| Johnny | Guitar Basics: Picking | 2020/05/30 12:40:32 | 3 | Content | Completed | 2 |
| Johnny | Guitar basics: Left hand technique | 2020/05/30 12:45:49 | 5 | Content | Started | 2 |
| Johnny | Guitar basics: Left hand technique | 2020/05/30 12:49:42 | 5 | Content | Completed | 2 |
| Johnny | Guitar Basics: Guitar parts | 2020/05/30 12:58:12 | 1 | Content | Started | 2 |
| Johnny | Guitar Basics: Posture | 2020/05/30 12:59:01 | 2 | Content | Started | 2 |
| Johnny | Guitar Basics: Picking | 2020/05/30 12:59:37 | 3 | Content | Started | 2 |
| Johnny | Guitar Basics: Picking | 2020/05/30 13:05:09 | 3 | Content | Completed | 2 |
| Mary | Chord diagrams | 2020/06/01 16:58:51 | 10 | Content | Started | 5 |
| Mary | Cowboy Chords | 2020/06/01 17:19:13 | 11 | Content | Started | 5 |
| Mary | Cowboy Chords | 2020/06/01 17:28:13 | 11 | Content | Completed | 5 |
| Mary | Recognizing Chords | 2020/06/01 17:30:00 | 12 | Assessment | Started | 5 |
| Mary | Recognizing Chords | 2020/06/01 17:36:28 | 12 | Assessment | Completed | 5 |
| Mary | Recognizing Power Chords | 2020/06/01 17:37:23 | 13 | Assessment | Started | 5 |
| Mary | Recognizing Power Chords | 2020/06/01 17:45:59 | 13 | Assessment | Completed | 5 |
| Mary | Recognizing Power Chords | 2020/06/01 17:47:39 | 13 | Assessment | Started | 5 |
| Mary | Recognizing Chords | 2020/06/02 09:10:29 | 12 | Assessment | Started | 6 |
| Mary | Recognizing Power Chords | 2020/06/02 09:12:22 | 13 | Assessment | Started | 6 |
| Mary | Recognizing Power Chords | 2020/06/02 09:15:41 | 13 | Assessment | Started | 6 |
| Mary | Recognizing Power Chords | 2020/06/02 09:18:41 | 13 | Assessment | Completed | 6 |
| ... | ... | ... | ... | ... | ... | ... |

[1] A typical event log; it should be noted that the "Resource" column is commonly used as an activity in process mining; however, the amount of possible resources and freedom of navigation that MOOCs give their students produces too many different paths when trying to discover a process model from it.

This approach is intended to examine the learning dynamics in student work sessions. It is necessary to identify the session that each interaction with a resource belongs to as a result. Certain MOOC systems include session definition and identification built in. However, this definition is not explicitly given in most MOOC systems, although it can be determined. For example, two successive interactions may belong to different sessions if a sufficient amount of time has passed between their timestamps. Ref. [30] reviewed several thresholds and their implications. After determining the session, the framework defines the case ID of the low-level log as a pair (user ID, session ID), implying that different sessions of the same student correspond to different cases in the log.

### 3.3. Stage 3: High-Level Log

The low-level event log obtained in stage two is similar to the input that a non-expert user in process mining would normally use in a tool such as Disco or ProM. However, due to the large number of resources and variants produced by this type of log, obtaining

process-driven insights is a challenging task. As a consequence, the framework's third stage describes seven high-level activities, each of which represents a distinct learning dynamic that reflects learner behavior independent of the resources consumed. We identify four dynamics relating to content consumption, or as [31] defines it, *"content related activities"*, and three dynamics involving interaction with evaluations, or *"graded assessment activities"* according to [31].

- **Progressing** : This shows the learning dynamic of a student who consumes a resource and then moves on to the next resource in the course in the right order. In other words, there is an interaction with a content-type resource that has either not been interacted with or in which the interaction was not previously completed; this interaction is then completed, and the student subsequently moves on to interact with the next resource, in the correct order. This dynamic recognizes a succession of behaviors similar to those described in [4,6] as *"Content Access"*, in [32] as *"V-V sequences"*, and *"Study"* in [33], but only on content that has begun and concluded in the same session and for the first time in the course.

- **Exploring**: This illustrates the learning dynamic of a student who engages with new content in a shallow fashion just to know what to expect, for example, to calculate the time required to study that content. In other words, an interaction is initiated but not completed with a content-type resource that has not previously been interacted with or with which the engagement has not previously been completed. This dynamic detects actions similar to those defined in [4,6] as *"Search"*, *"Content Access"*, or *"Overview"*; or *"Skipping"* in [33], but only on content that has not been completed previously and has not been completed this time.

- **Echoing**: This portrays the learning dynamic of a student who consumes a resource and then moves on to the next resource in the correct order. However, this is for resources that have already been completed. A relevant example is a learner who decides to review content before taking an exam. In other words, there is an interaction with a content-type resource that has already been accessed; this new interaction is then done again, and the learner moves on to the next resource in the correct order. This dynamic recognizes activities similar to those specified as *"Content Revision"* in [4,6], *"V-V sequences"* in [32], or *"Rewatch"* in [33], but it considers an organized sequence of this sort of activity, where the contents begin and conclude in an orderly manner.

- **Fetching**: This depicts the learning dynamics of a learner interacting with a previously completed material, with or without completion, in any order. An excellent example would be a student who, after failing an assessment question, rewatches (parts of or all of) a certain video to get the correct answer. In other words, there is an encounter with a previously finished content-type resource that does not follow the designated course sequence. This dynamic takes into account behaviors that are comparable to *"Content revision"* and *"Search"* actions described in [4,6] or *"Rewatch"* in [33].

- **Assessing**: This shows the learning dynamic of a student who interacts with and completes a previously uncompleted assessment-type resource. A block of many assessment activities in a row are crushed into a single dynamic, regardless of their sequence. In other words, an interaction occurs with an assessment-type resource that has not previously been engaged with or whose previous engagement was not finished, and this interaction is then completed. This dynamic recognizes activities similar to those specified as *"Assessment"* in [4,6], *"Q-Q sequences"* in [32], and *"Taking quizzes"* in [31], but only for evaluations that had not previously been completed.

- **Retaking**: This shows the learning dynamic of a student who initiates and completes previously completed assessments. For example, consider a user who did not receive a sufficient score and decides to retry in order to improve on their earlier performance. In other words, there is an interaction with an assessment-type resource that has already been finished, and this new interaction is then completed again. This dynamic considers groups of the same actions as the previous dynamic but distinguishes whether the evaluation was previously carried out by the student.

- **Skimming**: This reflects a student's learning dynamic in which he or she initiates but does not complete interactions with evaluations. For example, the student might be reading the questions before taking an examination seriously, or he or she might be examining an assessment to figure out where he or she went wrong. In other words, there is contact with an assessment-type resource that has been started but not finished. This dynamic takes into account behaviors that are comparable to *"View the quiz"* and *"Refer to quiz answers"* described in [34] and *"Viewing quiz results"* described by [31].

Figure 5 depicts, in terms of a decision tree, how each low-level event is related to a learning dynamic. It should be emphasized that low-level sequential events connected with the same learning dynamic are aggregated inside the same activity, and checking if a resource was completed before is measured across all sessions. The first timestamp in the sequence is used to determine the beginning of the activity, and the last one is used to identify the end. This generates a new event log with significantly fewer activities, as illustrated by the example in Table 2.

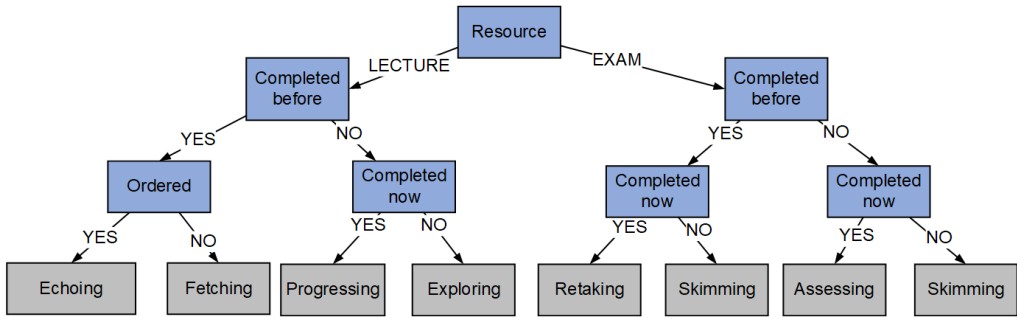

**Figure 5.** Criteria to assign to each high level activity.

**Table 2.** An example of an abstracted event log created using Table 1 data.

| User | Session | Activity | Started | Completed |
|---|---|---|---|---|
| Johnny | 1 | Exploring | 2020/05/28 13:05:10 | 2020/05/28 13:06:30 |
| Johnny | 2 | Exploring | 2020/05/30 12:30:22 | 2020/05/30 12:30:22 |
| Johnny | 2 | Progressing | 2020/05/30 12:30:22 | 2020/05/30 12:40:32 |
| Johnny | 2 | Progressing | 2020/05/30 12:45:49 | 2020/05/30 12:49:42 |
| Johnny | 2 | Fetching | 2020/05/30 12:58:12 | 2020/05/30 12:59:01 |
| Johnny | 2 | Echoing | 2020/05/30 12:59:37 | 2020/05/30 13:05:09 |
| Mary | 5 | Fetching | 2020/06/01 16:58:51 | 2020/06/01 16:58:51 |
| Mary | 5 | Echoing | 2020/06/01 17:19:13 | 2020/06/01 17:28:13 |
| Mary | 5 | Assessing | 2020/06/01 17:30:00 | 2020/06/01 17:45:59 |
| Mary | 5 | Skimming | 2020/06/01 17:47:39 | 2020/06/01 17:47:39 |
| Mary | 6 | Skimming | 2020/06/02 09:10:29 | 2020/06/02 09:12:22 |
| Mary | 6 | Retaking | 2020/06/02 09:15:41 | 2020/06/02 09:18:41 |
| ... | ... | ... | ... | ... |

[1] We observe that user "Johnny" has two sessions. Because the first session only begins and does not finish the content, session 1 would only include the Exploring dynamic. In their second session, however, it is observed that he progresses through resources 1, 2, and 3, which is the behavior of the Progressing dynamic. Then he breaks the course sequence to complete resource 5, which results in the creation of a new Progressing activity. Later, he revises resources 1 and 2 without finishing them, which is the Fetching dynamic, before finally starting and finishing resource 3 for the second time, so he ends up performing the Echoing activity. Meanwhile, user "Mary" starts her fifth session by looking over content 10, then completes resource 11, performing the behaviors of the Fetching and Echoing dynamics, respectively. She then starts and completes assessments 12 and 13, doing the Assessing dynamic, and ends the session by glancing over assessment 13, corresponding to the Skimming dynamic. The user checks assessments 12 and 13 again during her next session without completing them. This is equivalent to Skimming again, and then repeats assessment 13, finishing it this time, which is abstracted as the Retaking dynamic.

## 4. Learning Session Classification

We use cluster analysis with the K-means clustering method [35] and an Euclidean distance metric to distinguish differences between sessions and identify defining characteristics. The number of times each of the seven high-level learning dynamics appears is used as input for each session. Euclidean distance was selected in order to leverage existing tools and avoid implementation errors. Although studies have shown that distance metrics such as Manhattan [36] and Canberra [37] produce better results, determining which metric is superior in a dataset of this type is beyond the scope of this work.

To determine the number of clusters, we employ two k-means techniques: the elbow method and the silhouette method [38]. Following that, we calculate the z-scores of each of the clusters in every dimension to visualize their differences.

The metrics that were left out of the clustering method, such as the duration of each session and the quintile in which they are placed, as well as the z-score values obtained, are used to characterize the traits that set the discovered clusters apart. It is significant to note that because this study focuses on MOOC sessions rather than students, factors that describe the user who completes a session are not included in the clustering method. Nevertheless, unusual session patterns are excluded from the process. We specifically aim to rule out patterns that represent a user's unique behavior that is not reflected in any other instances.

## 5. Case Study: Successful Student Sessions in Coursera

A case study was conducted utilizing data from the Coursera platform's "Introduction to Programming in Python" course to demonstrate the framework's application and confirm its applicability with real data. The case study's goal was to investigate the learning dynamics that occurred in the sessions of students who successfully completed the course. We aim to address three research questions, which were first presented in Section 1.

Based on this, the following sections are presented: first, a description of the course and the application of the three levels of the framework related to the case study; second, the results obtained for the proposed research questions; and third, a discussion of the implications of these results and how they compare to other research on the subject.

### 5.1. Case Study & Framework

This study takes into account data gathered from a Coursera course held between 23 June 2017 and 14 April 2018. The course required a total of 17 h of time commitment and was divided into 6 modules, one for each week. In this research, 58 potential resources for interaction were considered, including 35 video lectures and 23 assessments. The decision to use this course for the case study was made for two reasons: first, ethics clearance had already been obtained on the data, as this is a secondary analysis of data previously used, and second, it was thought necessary to use pre-COVID-19 data so that it was not skewed by changes in MOOC use as a result of the pandemic.

15,420 people interacted with the platform during the observation period, of which 13,861 started the course during this time frame; we only keep the data of those who finished the course. Data was utilized in compliance with Coursera's Terms and Conditions of use for research in learning analytics and additional ethics clearance was provided by Pontificia Universidad Católica de Chile. Prior to being provided to the researchers, the data was anonymized, so it was impossible to individualize particular users.

The initial step in implementing the framework was to match Coursera's simple data model with the minimum data model proposed here. The Coursera data model had almost 75 tables. The Course Progress table was the most significant for this study since it tracked the course ID, the resource interacted with, the user who performed the interaction, the state (start/complete), and the date specifying when it occurred. However, because this table only contained IDs, it needed to be supplemented with the course information tables (Course Item Types, Course Items, Course Lessons, Course Modules, and Course Progress State Types) in order to establish the order of the resources within the course and obtain

contextual information about each resource. Table 3 shows the details of the fields in each of the tables used for the framework's application.

**Table 3.** Fields of the data tables used for the case study.

| Table Name | Field Name | Data Type | Description |
|---|---|---|---|
| Course Progress | course_id | ID | ID of the course in which an interaction occur. Foreign key to uniquely identify the course. |
| | course_item_id | ID | ID of the resource queried in the interaction. Foreign key to uniquely identify the resource. |
| | user_id | ID | ID of the user who makes the interaction. Foreign key to uniquely identify the user. |
| | course_progress_state_type_id | ID | ID of the state type of the interaction performed. Foreign key to uniquely identify the type of interaction performed. |
| | course_progress_ts | Timestamp | Moment in which the interaction is performed. |
| Course Item Types | course_item_type_id | ID | ID to uniquely identify a type of item. In this case there were only three different types of items but Coursera defines 19 possible ones. |
| | course_item_type_desc | String | Name of the resource type. |
| | course_item_type_category | String | Category to which the resource type belongs. Only items from the lecture and quiz categories were used. |
| | course_item_type_graded | Boolean | Boolean indicating whether or not the resource type has a grade associated with it. |
| Course Items | course_id | ID | ID of the course in which an item is placed. Foreign key to uniquely identify the course. |
| | course_item_id | ID | ID to uniquely identify a resource. |
| | course_lesson_id | ID | ID of the lesson in which the item is placed. Foreign key to uniquely identify a lesson. |
| | course_item_order | Integer | Integer to indicate the order of each resource within the lesson. |
| | course_item_type_id | ID | ID of the type to which the resource belongs. Foreign key to uniquely identify the type of resource. |
| | course_item_name | String | Name of the resource, written in Spanish originally. |
| | course_item_optional | Boolean | Boolean indicating whether or not the item if optional. |
| Course Lessons | course_id | ID | ID of the course in which a lesson is placed. Foreign key to uniquely identify the course. |
| | course_lesson_id | ID | ID to uniquely identify a lesson. |
| | course_module_id | ID | ID of the module in which the lesson is placed. Foreign key to uniquely identify the lesson. |
| | course_lesson_order | Integer | Integer to indicate the order of each resource within the module. |
| | course_lesson_name | String | Name of the lesson, written in Spanish originally. |
| Course Modules | course_id | ID | ID of the course in which a module is placed. Foreign key to uniquely identify the course. |
| | course_module_id | ID | ID to uniquely identify a module. |
| | course_module_order | Integer | Integer to indicate the order of each module within the course. |
| | course_module_name | String | Name of the module, written in Spanish originally. |
| | course_module_desc | Strings | A description of the module's contents and learning objectives. Written in Spanish originally. |
| Course Progress State Types | course_progress_state_type_id | ID | ID to uniquely identify a state type. |
| | course_progress_state_type_desc | String | "Name of the state. The table has only two states: 1 for "started" and 2 for "completed". |

[1] It's worth mentioning that the Course Progress table is the most significant for building the low-level event log, while the others are intended to supplement the log. The Course Items, Course Lessons, and Course Modules tables are used to determine course order and resource names. Course Item Types table is used to differentiate between assessment, lectures, and supplementary material (the latter was not included in the case study), while the Course Progress State Type table is only intended to differentiate between "started" and "completed" states.

The second step to apply the framework was to build the low-level event log, including the concept of session ID. To achieve so, an action was regarded to occur in the same session

as the previous activity if there was a lapse equal to or less that 60 min between them. This was determined using [30] maximum value of time-on-task. As a result, the case ID for the low-level event log was assigned as the pair (user ID, session ID). Our study only considered users who started and completed the course during the observation period. The completion of one of the two end-of-course tests determines if a user has finished the course. This results in a total of 320,421 low-level events. Furthermore, Coursera records progress through each question within an assessment as if it were a new event; for example, a student finishing an evaluation with 10 questions results in 11 events started and 1 completed. This duplication was later reduced, yielding a low-level event log of 39,650 events, 209 users for study, and a total of 7087 sessions.

Each event at this stage is made up of seven attributes, as exemplified in Table 1 : the user performing the event, the resource viewed, the timestamp of when this interaction occurs, the order of the resource in the progression (which is used to determine if the progression is sequential or chaotic), the type of the resource (to determine if it is a lecture or assessment), the state of the interaction (started or completed), and the session identifier to determine in which session this interaction occurs.

As a final step in applying the framework, we build the high-level event log by consolidating sequences of events into each of the seven high-level learning dynamics, using the criteria depicted in Figure 5. The resulting event log contained 18,029 events. When compared to the low-level log, this showed a 54.5% reduction in activities. The case ID for the high-level event log was established in the same way as it was for the low-level log (user ID, session ID). From the 7087 sessions and 209 users this yielded 1237 process variants. It should be noted that the five most common cases were associated with single activity sessions and accounted for 51.3% of all cases, with Skimming accounting for 26.7% of all cases.

### 5.2. RQ1: What Are the Characteristics of the Sessions That Involve Learning Dynamics in Which a Resource Is Revisited?

Our study discovered variations in learning dynamics when a resource is revisited, namely Echoing, Fetching, and Retaking. An exploratory analysis of the sessions using the Disco software revealed that Fetching appears in 13% of the cases, and that this activity appears to be significantly tied to the dynamics associated with assessment (i.e., Assessing, Skimming, Retaking); In 54.3% of cases when this action appears, it is preceded by one of the assessment-related activities; and in 50.9% of cases, Fetching is followed by some type of assessment.

Figure 6 depicts the process map of the fetching-related sessions. In this scenario, the Fetching of a content implies a specific search-related activity, either in preparation for an assessment or in reaction to a specific element that surfaced in an assessment and about which a specific doubt should be clarified. However, when the sessions included an Assessing or Retaking activity in addition to Fetching, the proportion shifted, with 25.6% executing the Fetching dynamic prior to the evaluation and 21.5% afterwards.

An examination of the material associated with Fetching revealed that the most frequently fetched resources were *2.2.2 Input, 3.1.1 If/Else , 3.2.2 For, 2.1.1. Data Types* (which makes sense because they are the basic building blocks required to create programs), and *6.1.4 List Functions*.

In sessions where the Echoing dynamic appears, students' behavior shifts. As illustrated in Figure 7, in most cases, this activity is related to Progressing, to the point that it is the activity that precedes or follows Echoing in 35.9% of the cases. This demonstrates that the student was oriented towards learning, and that questions developed during these study sessions, necessitating an in-depth examination of previously studied subjects. This is in stark contrast to Fetching sessions, which appear to be more directly tied to evaluative activities. In addition, there was also a link with the assessment dynamics in Echoing sessions. However, they differed in the order in which they appeared. This is because content repetition happened more frequently before the assessment as compared to the

versions that contained Fetching. In contrast, content review took place more frequently after the assessment dynamic occurred. Later, when reviewing the material related to Echoing, the contents of the second (Basic Python) and fifth weeks (Strings) were the most frequently repeated.

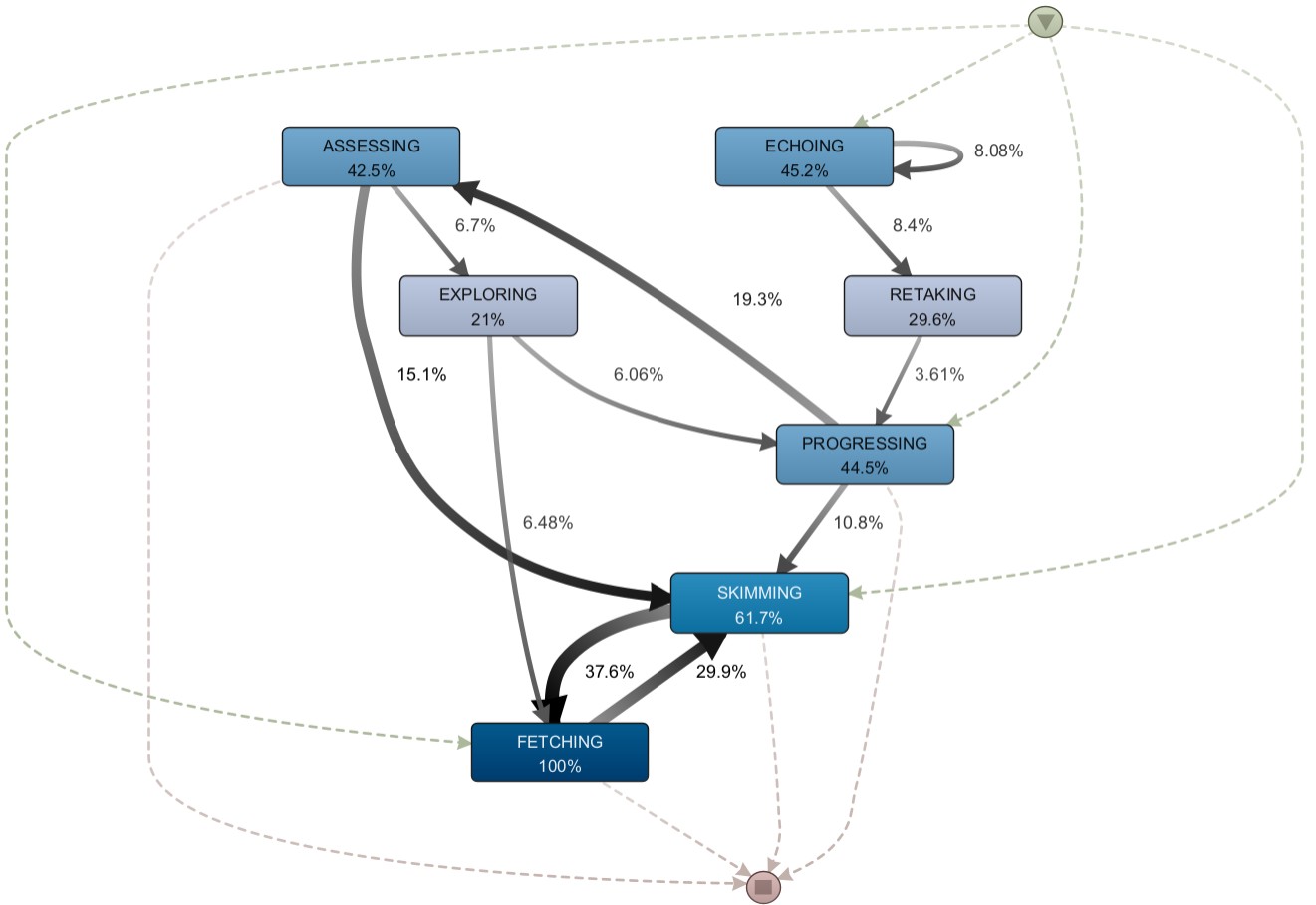

**Figure 6.** Fetching sessions.

Finally, Figure 8 depicts the behavior of the Retaking dynamic; in this case, it can be assumed that the user performs a session solely for the purpose of repeating assessments. This is due to the most common variant (25.9% of cases) involves repeating assessments and then immediately concluding the session. Similarly, the activity most closely associated with Retaking is Skimming, which denotes a dynamic in which students perform an assessment and then examine their results or consult their prior errors in order to improve their scores before a new attempt. Retaking to Skimming occurred in 33.6% of cases where this learning dynamic was present, while the reverse happened in 32.1% of cases. This means that one (or both) of these interactions occurred in 43.0% of cases with Retaking. When the most frequently repeated retaking-related evaluations were reviewed, one assessment in particular was shown to have a much higher number of Retaking activities than the rest (597 times out of a mean of 285). This assessment, which covered variables and input/output, contained a bug in one of the questions. The bug was later fixed after the observation date was recorded in the log.

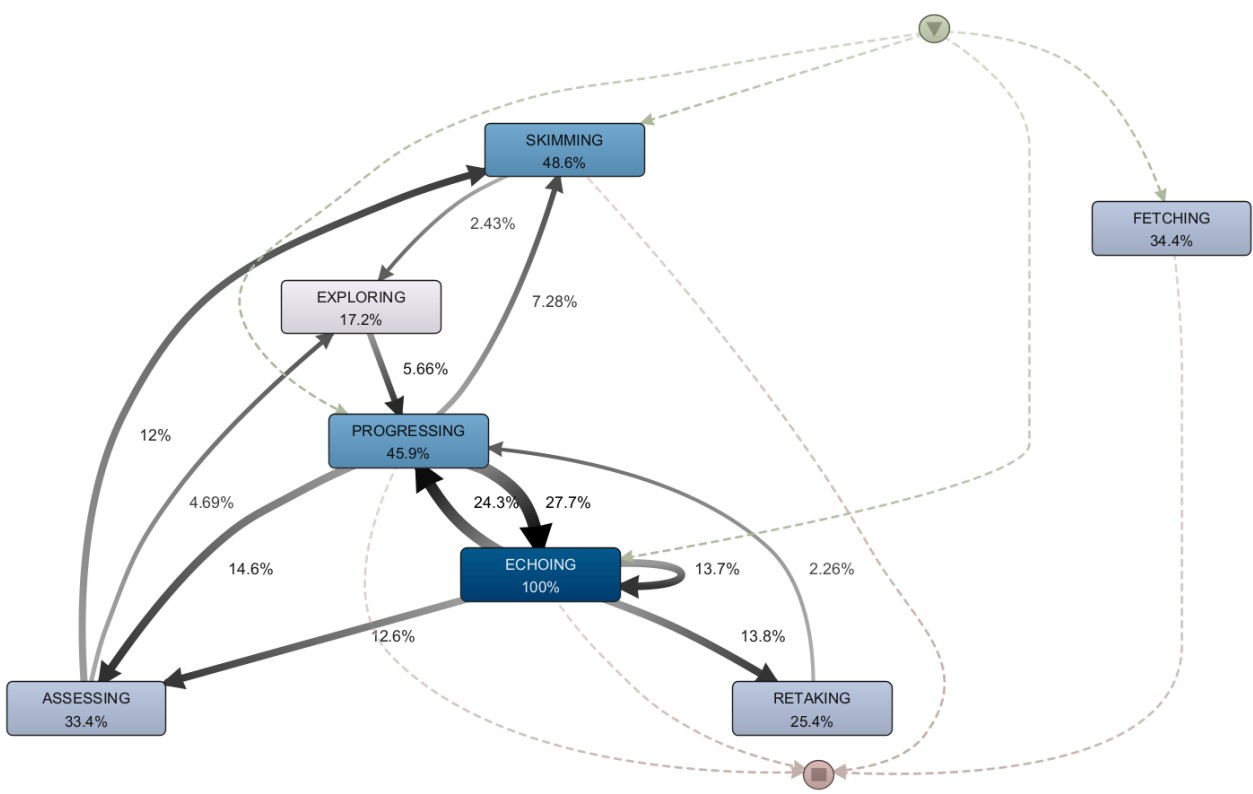

**Figure 7.** Echoing sessions.

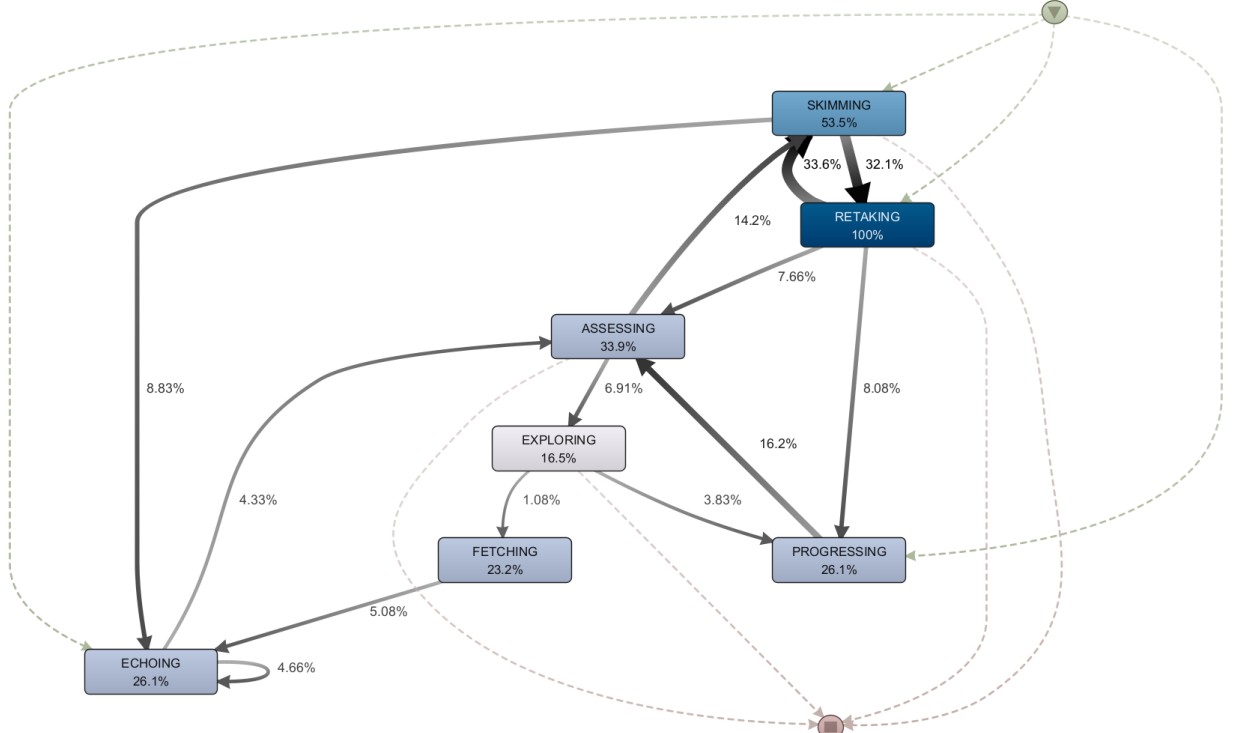

**Figure 8.** Retaking sessions.

*5.3. RQ2: Are There Differences in Terms of Learning Dynamics between the First and Final Sessions Carried Out by Students?*

Although analyzing the activities performed in a session is useful, it is also necessary to determine whether there are differences in the sessions of the students as they progress through the course. To do so, we grouped each session into quintiles based on the total number of sessions each student completed. As a result, the first and last quintiles were compared. This allowed us to confirm the existence of behavioral variations at the start and end of the course. Table 4 shows several statistics that characterize each quintile.

**Table 4.** Statistics on sessions by quintile .

|  | Q1 | Q2 | Q3 | Q4 | Q5 |
|---|---|---|---|---|---|
| % of cases | 19% | 20% | 19% | 19% | 22% |
| % of events | 23% | 23% | 20% | 18% | 16% |
| N° of cases | 1366 | 1429 | 1371 | 1359 | 1562 |
| N° of events | 4120 | 4097 | 3655 | 3303 | 2854 |
| Mean case duration | 49 min | 46.9 min | 38.3 min | 36 min | 28.9 min |
| Median case durations | 28 min | 16.5 min | 8.9 min | 5.8 min | 0 min |
| Variants | 446 | 395 | 336 | 298 | 228 |
| Most common variant | Progressing (18.23%) | Skimming (21.27%) | Skimming (27.64%) | Skimming (32.52%) | Skimming (40.4%) |
| Mean activities per case | 3.02 | 2.81 | 2.61 | 2.43 | 1.83 |
| Median activities per case | 2 | 2 | 1 | 1 | 1 |

The process model of the initial sessions is depicted in Figure 9, from which it can be determined that the most prevalent activities are related to the dynamics of orderly and thorough learning (Progressing 63.9% and Echoing 29.7%). Furthermore, a rather low commitment to assessment can be detected at this stage in the course, as students were seen undertaking sessions without completing an assessment in 66.3% of cases, despite the Skimming activity appearing in 36.7% of cases. This notion is reinforced by the fact that the progressing activity appears more than once in 5.78% of the cases. This implies that the students opted not to disrupt their study process by bypassing the assessments that came in between video lectures. We expected the Progressing and Exploring activities to be more frequent at the start of the course and to decrease as the course progressed because they correspond to the first time a piece of content is viewed. It is worth noting, however, that the frequency of Echoing activity was 29.7% during the initial sessions. This is relatively high given that the students have not yet completed all of the course content.

Furthermore, by categorizing the sessions into quintiles, it was possible to illustrate that the sessions at the start of the course had higher changes in terms of learning dynamics. Despite accounting for only 19% of the sessions, the initial ones were shown to account for 23% of all events in the high-level log. The number of events in each quintile decreased as the process was repeated, with the last quintile accounting for only 15% of the total events in the high-level event log.

Examining the process map of the course's final sessions, as shown in Figure 10, reveals that these are primarily related to assessment activities. This is because the dynamics of Skimming, Assessing, and Retaking appear more frequently than the dynamics associated with content. For instance, 39.9% of the former engaged in at least one Assessing or Retaking activity. However, it is noteworthy that 40.4% of cases belonged to students who just completed Skimming and then ended the session. This reveals that a considerable proportion of students only signed up to explore the questions without completing the overall evaluation. In terms of content dynamics, the Progressing activity was the one that typically started the sessions in which it featured, and it was usually followed by assessment activities, particularly Assessing. This reflects variations from the initial stages of the course, when the user preferred to study or repeat topics more frequently.

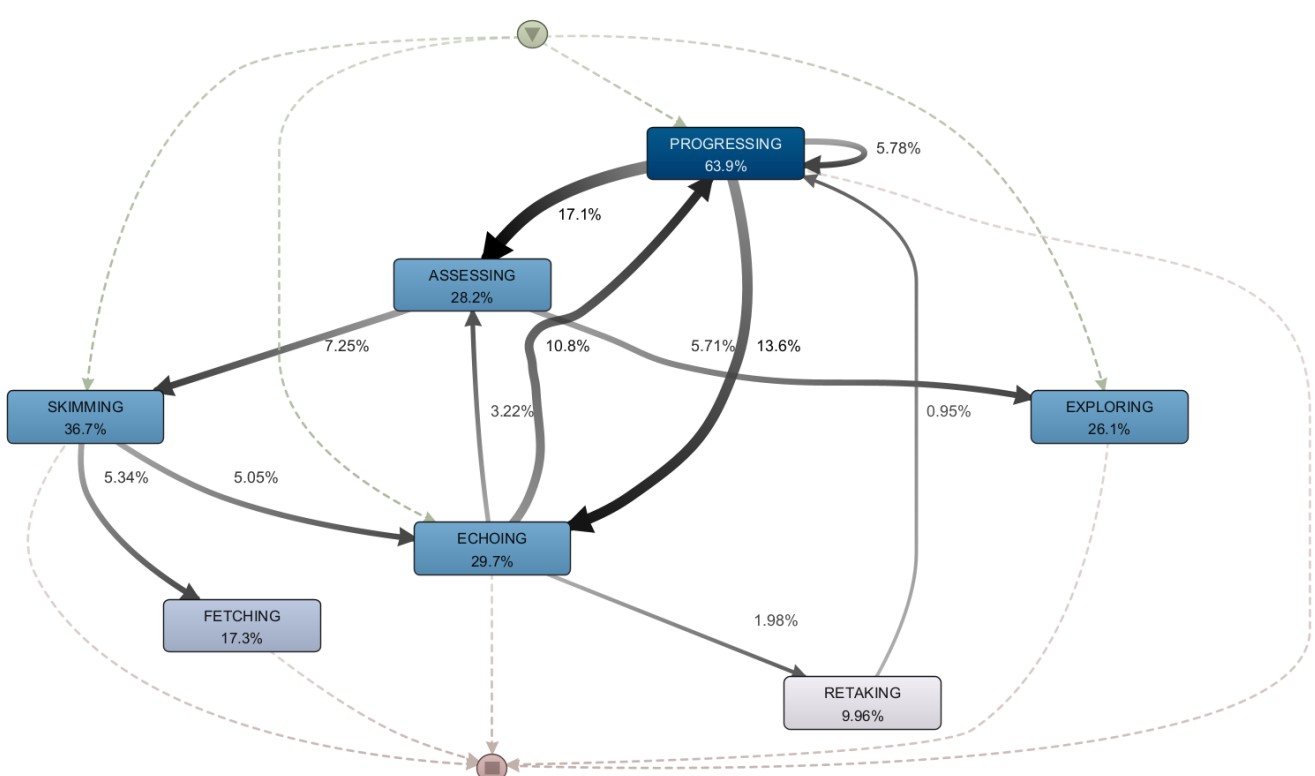

**Figure 9.** Initial sessions, or the course's first quintile.

Furthermore, it should be noted that even at the end of the course, the Progressing activity appeared more frequently than the Echoing activity and for a longer average time (23.5 minutes versus 9.6 minutes). Similarly, the practice of continuing to study while missing an assessment significantly reduced its incidence, accounting for only 2 incidents, or 0.1% of these sessions.

### 5.4. RQ3: What Types of Mooc Sessions Do Successful Students Go Through and How Do They Differ from Each Other?

Using the results from the processes outlined in Section 4 for determining the number of clusters, the number that best segregated the data was deemed to be four. The duration, average number of activities, and number of cases classified in each cluster are shown in Table 5, while the results for each variable after z-score normalization are shown in Figure 11. Each cluster's differences and distinctions are detailed below. It is worth noting that two outliers were detected and deleted from the sample. These two cases in particular had the two longest sessions at 334 and 252 min, respectively, and the cases with the most activities per session at 89 and 73. Both sessions were completed in the middle of the course by the same user and described behavior similar, although exaggerated, to Cluster 3.

*Cluster 1* accounts for the majority of the behavior observed in the log, with relatively short (<20 min) sessions containing one or two activities. Given the small number of activities in each session, the z-score for each dimension is expected to be somewhat low. Consequently, the activities with the lowest z-scores in this cluster are Skimming ($-0.702$) and Retaking ($-0.653$). When determining the point in the course at which the cases of this cluster appear, it is striking that, despite the low z-score of activities associated with assessment, the sessions of cluster 1 are mostly located in the course's final quintile (25.92%), while the initial quintile only has 16.43%, and quintiles 2, 3, and 4 have values ranging from 18.68% (Q2) to 19.67% (Q4). Despite the fact that the majority of cases belong to this cluster, at event level, these account for just 42.13% of the reduced log.

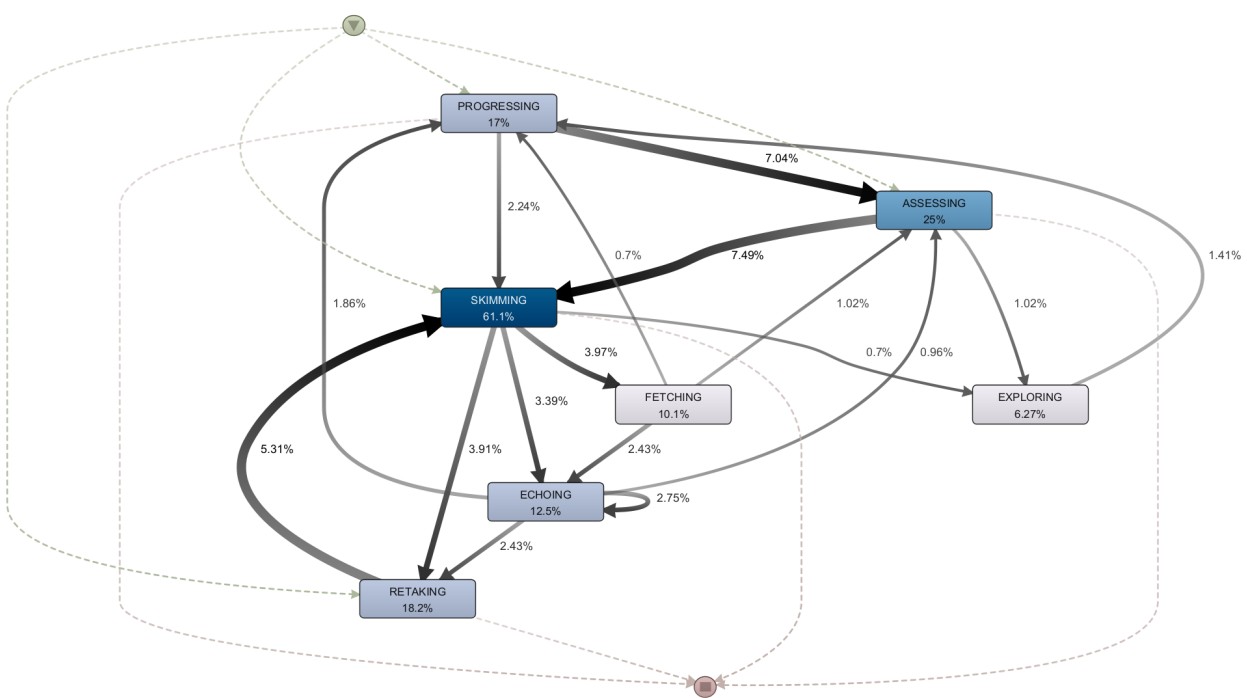

**Figure 10.** Sessions at the end of the course or at the fifth quintile.

**Table 5.** Cluster statistics.

| Cluster | N° of Sessions | Percentage | Average Time (Minutes) | Average Activities |
|---|---|---|---|---|
| Cluster 1 | 5288 | 74.6% | 19.4 | 1.4 |
| Cluster 2 | 435 | 6.1% | 108.9 | 8.2 |
| Cluster 3 | 33 | 0.5% | 163.5 | 26.2 |
| Cluster 4 | 1329 | 18.8% | 93.8 | 4.42 |

*Cluster 2* groups about 6% of the instances, however the events in it account for 20.12% of the total events in the log. This cluster is distinguished by longer sessions (about 108 minutes) and a larger number of activities per session (an average of 8.267 each). In this instance, four dynamics had high z-scores: Assessing (0.315), Exploring (0.518), Echoing (1.509), and Fetching (1.42), with the latter two having the highest scores for their respective activity. It is worth noting that in this situation, the majority of the sessions grouped belong to the middle of the course, with the bulk of them falling into the second quintile (25.75%), third quintile (21.84%), and fourth quintile (19.77%). Sessions in this cluster tend to occur less frequently in initial sessions (17.01%) and final sessions (15.63%). The high z-score of activities involving content repetition, paired with their location in the middle of the course, may imply that these sessions represent an active effort by the student to overcome some barrier, such as learning a new subject or overcoming the difficulty of an assessment.

*Cluster 3* makes up 0.5% of all cases and 4.85% of all events. These sessions correspond to long-duration sessions that generally revolve around many interactions between the Retaking and Skimming activities, with the appearance of one of the other 5 learning dynamics in certain cases. As in Cluster 2, the sessions in this scenario take place mostly in the middle of the courseThis is due to 36.36% occurring in the second quintile, 27.27% in the third quintile, and 18.18% in the fourth quintile. With only five cases at the beginning (15.15%) and one at the end of the course.

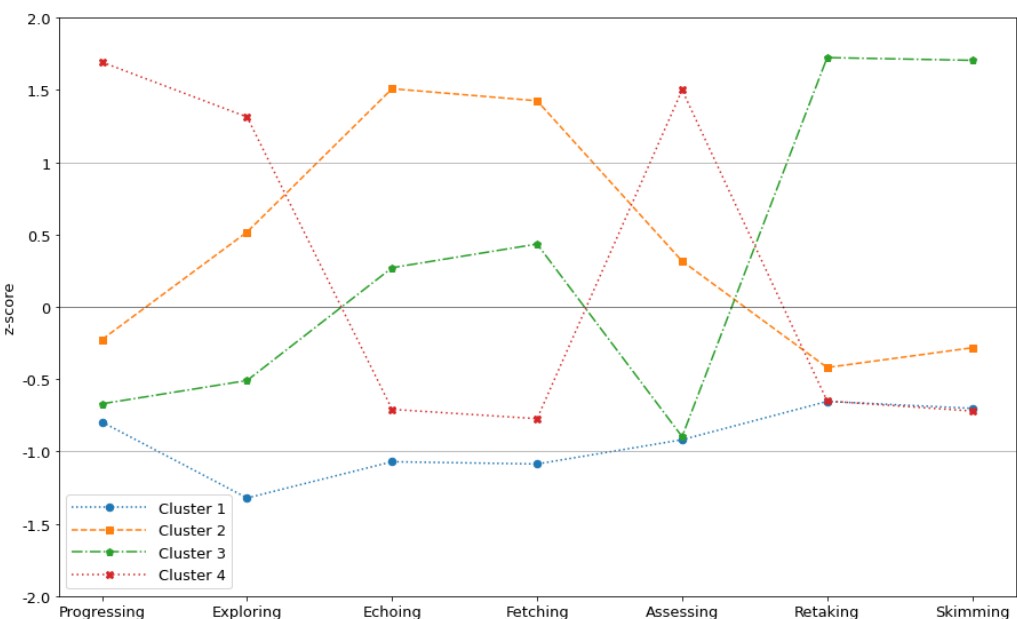

**Figure 11.** Z-scores of the amount of activities for each cluster.

*Cluster 4* is the second largest cluster in terms of size, accounting for 18.88% of all sessions and 33.89% of the events in the high-level event log. The z-score of the progressing activity in this cluster is the highest in the sample, at 1.692. This means that this cluster identifies sessions in which the learner is progressing sequentially and thoroughly following the course program. This is supported by the fact that the Assessing activity, that is, performing assessments for the first time, also had the highest Z-score in the sample at this point, with a value of 1.500. When the distribution of sessions is compared to the quintiles of course progression where they are situated, the majority of the sessions in this cluster appear at the start of the course (31.45%). Then, as the course progresses, the number of sessions of this type gradually decreases, with only 9.17% of them coming from the course's final quintile. The fact that only 4.42 activities are seen on average during each of these sessions, which last an average of 93.8 minutes, is also interesting to note. This could indicate an intention to advance through the course, taking into account both assessments and video lectures, in order to successfully complete it.

*5.5. Discussion*

First and foremost, the findings indicate that students' behavior evolved throughout the course, as they were initially hesitant to evaluate themselves and preferred to view the content rather than assess their level of knowledge. The situation changed as participants progressed through the course, with 25% of the total number of events reported in the log resulting in completed assessments, either for the first time or by redoing a previously completed assessment. This shows that students' dedication to the subject grew as the course continued.

According to the case analysis performed, the most frequent variants were of a single activity, and the longest sequences typically featured activities of the same type (e.g., Progressing and Echoing or Skimming and Retaking) indicating a student's purpose when carrying out a MOOC session. This reinforces [8] findings that effective learners typically focus their session time on a single learning resource (assessments, lectures, etc.), with the focus shifting from session to session. It is is also consistent with [39] research on study patterns, where it was discovered that a sizeable portion of learners in the MOOC focus their time on a specific task, such as viewing videos or submitting an assignment, and with [5,21], which recognizes that the most common interaction patterns are associated with the use of a single type of resource (video-lecture only and assessment-summative only, respectively). Additionally, this is in line with the most common learning tactics

reported in [4,6], where content-only and assessment-only tactics account for a considerable portion of the presented sample. Our research builds on these findings by distinguishing between first-time and re-engagement with content and assessments.

Excluding students who dropped out of the course from the analysis allowed for the description of behaviors that successful students commonly display. From the five clusters obtained, four types of sessions were identified: Short sessions with a specific goal (Cluster 1), long sessions in which the student moves through the course sequence in an orderly fashion (Cluster 4), sessions associated with Retaking-Skimming dynamics (Cluster 3), and sessions with content search and repetition (Cluster 2). The clusters obtained are akin to those reported in [8], except for the two associated with No activity and Drop out, which are not present in our analysis. Furthermore, the patterns reported here are aligned with the activity patterns outlined in [34] for successful students and in [19] for the cluster researchers named *"Achievers"*.

One of the unexpected findings in this study was that the Skimming dynamic was the most common variant, accounting for 26.7% of the high-level log. This could indicate the need to refine this dynamic because the review of an unfinished assessment could be the result of several factors, including: reviewing the difficulty of the content to be assessed in preparation for a serious attempt to complete it; reviewing mistakes made in previous attempts; and using the questions as learning examples, among others. Particularly in a programming course, this could be related to the necessity to check the code's syntax or to better understand some control flow structure or data type. It is worth looking into the possibilities of obtaining more precise information from MOOCs in order to determine the actual objectives behind these unfinished assessment attempts. One method could be to investigate how the variants that incorporate Skimming connect to the content being viewed. Another approach is to contrast with external data, such as questionnaires, to uncover patterns underlying the skimming activities in order to describe more precise dynamics.

This research illustrates how a domain knowledge-based abstraction enables the description of student–MOOC interactions that would not be possible without this transition from low to high level. However, because our research cannot exist in a vacuum, it is necessary to enhance the findings of this work with advancements from other researchers. First, the framework aims to determine students' intentions from the available data (for instance, progressing versus exploring). However, similar to what has been done in other MOOC studies such as [8], such extrapolations could be improved if the framework were supplemented with certain additional instruments, for example, surveys and interviews. Second, dealing with fine- and mixed-grained events is a challenging task, as stated in [22]. This work serves as an example of how this issue is not unrelated to data from educational systems. It also emphasizes how crucial it is to establish specific methods for this domain and evaluate the effectiveness of techniques provided in other domains on this sort of data. In this regard, rather than competing approaches, domain-driven event abstractions and data-driven event abstractions should be considered as complementary tools that can support each other. With this in mind, our framework could be supplemented with some of the data-driven event abstraction techniques described in [13], creating a hybrid method that combines the two approaches with the aim of improving the dynamics suggested here and delivering a high-level event log consistent with both the source data and prior knowledge of the process. Finally, to validate the framework's universality and utility, it must be evaluated using data from various MOOCs and platforms. Similarly, different courses would allow assessing their extensibility in relation to interactions not covered in this case study, such as forums or projects.

## 6. Conclusions

This study proposes a domain-driven event abstraction framework for creating a high-level event log to analyze learning dynamics in MOOC sessions. The framework, in specific, is divided into three stages: (1) the minimal data model required; (2) the creation

of a low-level event log; and (3) the classification into seven high-level activities that can be utilized to construct an abstracted high-level event log.

A case study that examined the learning dynamics of the sessions of students who completed the course successfully validated the application of the framework in a real-world setting. The examination of the dynamics of the resources that are reviewed again lets us identify problems in the course and reveals that most students start a study session with a certain goal in mind (advance in the course, complete assessments, clear up questions, etc.). Similar distinctions were made between student behaviors at the start of the course and those at the end, since early on, students prioritized content consumption over working on assessments. However, when involvement to the course increases, students begin to actively participate in evaluation activities as their interest in the subject grows. Finally, the sessions could be divided into 4 different clusters, highlighting that while the development of relatively short sessions with a specific goal is the most common behavior of students in MOOCs, there are also sessions of extended duration in which students invest more time in the course by either progressing through it, attempting to improve their assessment results, or navigating through the course and revisiting content and assessments.

Among the limitations of this study is the use of sessions as the unit of analysis in our proposed event abstraction; thus, reviewing lower-level behavior patterns, such as behavior within an assessment, or higher-level ones, such as behavior during a course week or an overview of the entire course, requires their own domain-driven abstractions, while the framework was intended to be expanded and modified, as previously said, it begins with a separation between two basic categories of resources: content and assessment. This means that applying similar learning dynamics to courses that do not follow this order, such as project-based courses, may not make sense. However, the proposed abstraction approach would still be a valuable alternative for carrying out process analysis without having to deal with the complexities of data generated by students with total freedom of navigation through MOOC resources. Similarly, while the concepts explored in this research might be useful in synchronous and hybrid courses, such as b-learning courses hosted in LMS systems like MOODLE, our approach does not take into account the possibility of activities that might exist outside the system and that are not recorded in the database. This restriction must be taken into consideration by any application in a setting where there is interaction with parties outside the system, such as in a classroom.

This research demonstrates that utilizing event abstraction techniques on educational data enables the discovery of outcomes that would not be obvious when using data mining or process discovery techniques directly. One potential route for expanding on this research would be to identify behaviors that necessitate student interactions, such as forums, group projects, or peer assessments, as their own learning dynamics. Another alternative is to compare how the top-down learning dynamics described are correlated with a bottom-up abstraction output. This might allow for the detection of previously unknown learning dynamics or indicate the necessity for intermediate abstraction layers between the low-level of resources viewed and the activities conducted by a student during a MOOC session. In a related vein, given that all interactions are mediated by a MOOC system, it would be interesting to study how cultural [40], usability [41], or other factors, such as users' special needs (e.g., autism [42]), affect study session behavior. Finally, emphasizing the usefulness of event abstractions in an educational context opens up the possibility of using this type of technique in other contexts and educational problems where digital traces exist and process mining has previously been used, such as dropout detection [17], curriculum analytics [43,44], quiz taking behavior [45], and so on.

In recent years, the interest in understanding the behaviors, actions, tactics, strategies, and dynamics of MOOC users has been increasing, so there is a compelling need to systematically consolidate and organize the findings on the subject obtained so far. This is emphasized by the fact that techniques from a variety of disciplines, such as epistemic networks [46,47], ordered network analysis [6], process discovery [7,15,16], sequence mining [17,48], n-gram analysis [32], and traditional machine learning [19,20], are currently

being employed to describe these phenomena. This highlights the importance of using a common framework and taxonomy to help to refine the scope of future research proposals on this topic.

**Author Contributions:** Conceptualization, J.M.-G. and L.H.; Data curation, L.H.; Formal analysis, J.M.-G. and L.H.; Funding acquisition, J.M.-G.; Methodology, J.M.-G.; Project administration, J.M.-G. Software, L.H.; Supervision, J.M.-G.; Validation, L.H. and J.M.-G. Visualization, L.H. and J.M.-G.; Writing—original draft, L.H. and J.M.-G. Writing—review and editing, L.H. and J.M.-G. All authors have read and agreed to the published version of the manuscript.

**Funding:** This work is partially supported by ANID FONDECYT 1220202, FONDEF IDeA I+D 2210048 and ANID-Subdirección de Capital Humano/Doctorado Nacional/2022-21220979.

**Institutional Review Board Statement:** Not applicable

**Informed Consent Statement:** Not applicable

**Data Availability Statement:** The data provided in this work are available from the corresponding author upon reasonable request. The data is not publicly available since its dissemination is under the supervision of Pontificia Universidad Católica de Chile

**Acknowledgments:** The authors would like to thank the editors and reviewers for their valuable remarks and recommendations that contributed to enhance the quality of this publication.

**Conflicts of Interest:** The authors declare no conflict of interest.

## Abbreviations

The following abbreviations are used in this manuscript:

| | |
|---|---|
| B-learning | Blended Learning |
| ID | Identifier |
| LMS | Learning Management System |
| MOOC | Massive Open Online Courses |
| MOODLE | Modular Object-Oriented Dynamic Learning Environment |
| RQ | Research Question |

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
