# Peer review of "A Domain-Driven Framework to Analyze Learning Dynamics in MOOCs through Event Abstraction"

_applsci, doi:10.3390/app13053039_

Round 1

Reviewer 1 Report

The numbering of citations includes errors

There are several typos ….no spellcheck has been used…

The Dataset need to be defined

 The data features should be described in detail

 You need to introduce the metadata structure of the feature and distribution of the dataset

 It seems that the article does not respect the "fair use" aspects of copyright. One can copy limited text from a published work provided that it is duly cited and referenced via a ftnt or end note and then also in the Bibliography.

Author Response

Dear Reviewer:

First and foremost, we would like to thank you for your feedback, which assisted us in improving the document's quality.
Due to some overlapping between comments and to avoid inconsistencies, we produced an unified answer for all reviewers.
Please see the attached file; the comments to your observations can be found in the Reviewer 1 section.

Kind regards.

Reviewer 2 Report

This article offers a domain-driven top-down method that allows educators who are unfamiliar with process analytics 

to search for a set of predefined high-level concepts in their own MOOC data, simplifying the use of typical process mining techniques.

My main strong points and concerns are as follow:

(-) Overall problem to be solved is well explained on "Introduction", however, when proposed framework is stated, only simplicity is argumented to justify the method. Can authors describe other possible alternatives and the disadvantages of them? (or de advantages of the chosen one)?

(-) Research questions are not justified (don't come from literature). I'm ok with that as long as the framework allows to future researchers modify these research questions. That should be clearly stated.

(+) Related work and process mining sections are well explained and referenced.

(+) The 7 dynamics are well explained and comes from a journal reference. Figure 5 helps understanding the criteria to assign each activity.

(-) It is not explained why K-means clustering method with Euclidean distance metric is used for the cluster analysis.

(-) My mayor concern is about the case study. It is more than 4 years old (do researcher have other different case studies? If yes, are results consistent?) Which are the characteristics of the course (guitar)? What about other courses that are not as "lineal" as learning to play an instrument? That is specially relevant in sections 5.2, 5.3 and 5.4 where results might be strongly dependant of the activities of the course and the nature of it.

(+) References are good in quantity and quality. They are very up to date (nearly all of them are from the last 5 years).

In summary, research is appropriate and can lead to a better understanding of MOOC courses. The case study should be taken only as a proof that can be applied instead of analysing in detail the results obtained on it.

Author Response

Dear Reviewer:

First and foremost, we would like to thank you for your feedback, which assisted us in improving the document's quality.
Due to some overlapping between comments and to avoid inconsistencies, we produced an unified answer for all reviewers.
Please see the attached file; the comments to your observations can be found in the Reviewer 2 section.

Kind regards.

Reviewer 3 Report

The biggest problem of your work is that you have a bright idea but you have not taken time to put it into form to make it easily understandable for potential readers. Education is a field where many decision makers lack technical knowledge - you really need to help them understand the significance of your work by following a clear structure and avoiding tech-heavy phrasing where possible. 

---

The problematic thing with your paper is the description of what you do. This problem is present in abstract where from rows 1 to 11 you are describing a general background of your study and then, unexpectedly, already start presenting results on row 12. In addition, in the sections 3 and 4 are difficult to understand where you are describing that what you do in this study or where you describing the relevant concepts in general. And more, in 5 you are presenting your research questions and results - without a clear description of your sample (how did you find them, were they informed and why not, what was their demographics, etc.) and your method (how did you plan your experiment; how did you collect the data, how did you process the data). You have described your framework but you have not sufficiently described how you tested it (used it in your study). These things can be self-explanatory to you but an interested reader could become really confused. 

A standard paper structure is: Introduction/Theoretical Background; Materials and Methods; Results; and Conclusions/Discussion. I would really recommend you to restructure your work, considering that standard. 

===

Some smaller issues:

Figure 1 is not in English (zoom up). Please refer to original sources of Figure 1 (citation needed). On Figure 1, indicate the source of raw data (what field is the data from).

On Figure 3, there is misspelling in the "Process mining analysis" block.

The research question 3 could be rephrased. Perhaps instead of "experience" a more suitable phrase could be "pass through" or something similar?

On rows 275-276 you mention that "unusual session patterns ... are excluded..." Please describe the criteria or process for determining unusual patterns. 

On rows 283-288 you again "define" your research questions. As these were already "proposed" on rows 71-75, in order to avoid confusion, you should choose where you present your research questions - or give an explit reference to the research questions that were already presented in section 1.

In 5.1 please describe if your users were informed about your study and if not, please give reasoning why not. 

In Discussion / Conclusions you have not discussed your paper's limitations and future perspectives of this field. 

Author Response

Dear Reviewer:

First and foremost, we would like to thank you for your feedback, which assisted us in improving the document's quality.
Due to some overlapping between comments and to avoid inconsistencies, we produced an unified answer for all reviewers.
Please see the attached file; the comments to your observations can be found in the Reviewer 3 section.

Kind regards.

Reviewer 4 Report

Presentation:

 • Quality of Figures: Figures are legible but as it is important part of study so need to be more clear. They support the information provided in the paper but add more information and provide proper referenced within the text. Study shows computer-generated figures which is encouraging.  • Quality of English language: The spelling, Sentences, grammar, word usage, punctuation, etc. of the text need to be scrutinized. Check it again, there are mistakes.  • Organization: The paper content and style used is presented in a manner that is easy for the reader to follow.  • Completeness: Appropriate citations in the reference list, descriptive captions on all figures and tables, and conclusions substantiated by statements within the previous text is presented clearly.       Demerits:  

1.      Structure your abstract as follows- 1) Background 2) Aim/Objective 3) Methodology 4) Results 5) Conclusion. Write 2-4 lines for each  and  merge everything in one paragraph without any subheading

2.      Abstract must contain the motivation and objective of the article. The Abstract must be very clear and the motive of the paper should be represented in a nutshell.

3.      Introduction should be of 5-7 solid paragraphs and provide structure of work at the end of the Introduction section.

4.      Add more contribution to your study field.

5.      The purpose of study not clear.

6.      Make highlight for objectives

7.      Remove any table or figure which is taken from web. Otherwise you have to get approval from publisher and author in a provided form by springer.

8.      Please avoid to write definitions of terms like Event Abstraction, MOOC, Learning Dynamics, Process Mining, etc., which are already available over web, try to cite work for such information.

9.      In summary, only provide useful content in your work.

10.   These are Title related to your area, you may use.

a. Alshar’e, M., Albadi, A., Jawarneh, M., Tahir, N., & Al Amri, M. (2022). Usability evaluation of educational games: an analysis of culture as a factor Affecting children’s educational attainment. Advances in Human-Computer Interaction2022.

b. Alshar’e, M., Mustafa, M., & Bsoul, Q. (2022). Evaluation of E-Learning Method as a Mean to Support Autistic Children Learning in Oman. Journal of Positive School Psychology6(3), 3040-3048.

c. Alshar’e, M., Albadi, A., Mustafa, M., Tahir, N., & Al Amri, M. (2022). Hybrid user evaluation methodology for remote evaluation: case study of educational games for children during covid-19 pandemic. Journal of Positive School Psychology6(3), 3049-3063.

d. Alshar’e, M., Albadi, A., Mustafa, M., Tahir, N., & Al Amri, M. (2022). A framework of the training module for untrained observers in usability evaluation motivated by COVID-19: enhancing the validity of usability evaluation for children’s educational games. Advances in Human-Computer Interaction2022, 1-11.

Author Response

Dear Reviewer:

First and foremost, we would like to thank you for your feedback, which assisted us in improving the document's quality.
Due to some overlapping between comments and to avoid inconsistencies, we produced an unified answer for all reviewers.
Please see the attached file; the comments to your observations can be found in the Reviewer 4 section.

Kind regards.

Round 2

Reviewer 1 Report

Accept

Reviewer 3 Report

Thank you for taking the suggestions into consideration.